# Exposure Assessment of Ambient PM2.5 Levels during a Sequence of Dust Episodes: A Case Study Coupling the WRF-Chem Model with GIS-Based Postprocessing

**DOI:** 10.3390/ijerph20085598

**Published:** 2023-04-20

**Authors:** Enrico Mancinelli, Elenio Avolio, Mauro Morichetti, Simone Virgili, Giorgio Passerini, Alessandra Chiappini, Fabio Grasso, Umberto Rizza

**Affiliations:** 1Department of Industrial Engineering and Mathematical Sciences, Università Politecnica delle Marche, 60131 Ancona, Italy; 2National Research Council—Institute of Atmospheric Sciences and Climate (CNR-ISAC), 88046 Lamezia Terme, Italy; 3National Research Council—Institute of Atmospheric Sciences and Climate (CNR-ISAC), 73100 Lecce, Italy

**Keywords:** regional chemical transport model, geographical information system, dust, dust alert system, risk prediction system, short-term exposure, air pollution, population exposure

## Abstract

A sequence of dust intrusions occurred from the Sahara Desert to the central Mediterranean in the second half of June 2021. This event was simulated by means of the Weather Research and Forecasting coupled with chemistry (WRF-Chem) regional chemical transport model (CTM). The population exposure to the dust surface PM2.5 was evaluated with the open-source quantum geographical information system (QGIS) by combining the output of the CTM with the resident population map of Italy. WRF-Chem analyses were compared with spaceborne aerosol observations derived from the Moderate Resolution Imaging Spectroradiometer (MODIS) and, for the PM2.5 surface dust concentration, with the Modern-Era Retrospective analysis for Research and Applications, Version 2 (MERRA-2) reanalysis. Considering the full-period (17–24 June) and area-averaged statistics, the WRF-Chem simulations showed a general underestimation for both the aerosol optical depth (AOD) and the PM2.5 surface dust concentration. The comparison of exposure classes calculated for Italy and its macro-regions showed that the dust sequence exposure varies with the location and entity of the resident population amount. The lowest exposure class (up to 5 µg m^−3^) had the highest percentage (38%) of the population of Italy and most of the population of north Italy, whereas more than a half of the population of central, south and insular Italy had been exposed to dust PM2.5 in the range of 15–25 µg m^−3^. The coupling of the WRF-Chem model with QGIS is a promising tool for the management of risks posed by extreme pollution and/or severe meteorological events. Specifically, the present methodology can also be applied for operational dust forecasting purposes, to deliver safety alarm messages to areas with the most exposed population.

## 1. Introduction

Saharan dust outbreaks impact air quality with peaks in particulate matter (PM) concentrations and implications on human health in the Mediterranean area. North Africa contributes to approximately 50% of the global dust loading [1], with higher quantities from western north Africa compared to eastern north Africa [2]. The largest fraction (60%) of the total emitted dust from Sahara remains in Africa, being transported in the Sahelian region along the so-called “meningitis belt” [3], which is recognised to be at high risk of epidemics of meningococcal but also pneumococcal meningitis. A relevant (10%) Saharan dust quantity is also transported across the Mediterranean Sea to Europe [4,5] following seasonal meteorological patterns [6,7].

The annual cycle of dust aerosols shows maxima at spring (April/May) and summer (June/July) over the Mediterranean, according to the analysis of satellite data retrieved for a 15-year period in the work by Gavrouzou et al. [8]. Mifka et al. [9] depicted the typical scenario resulting in highest dust deposition in the Adriatic Sea, with dust sources located in north-western Africa (Algeria, Libya, and Morocco) and synoptic conditions characterized by an upper-level trough and a deep cyclone over the central Mediterranean.

Calidonna et al. [10] observed a recurrent pattern for dust outbreak episodes in southern Italy, with trajectories originating in the Sahara Desert moving eastward to the dust source regions in central/north-eastern Africa. A large latitudinal gradient has been reported during the desert dust episodes at ground levels [11,12], both in terms of dust contribution to mean PM10 annual levels [12], and of annual average dust AOD [13,14], considering the southern/central/northern Italian peninsula. In Italy, desert dust contributed between 0.7 and 1.8 µg m^−3^ to the PM10 yearly average in a seven-year period, with approximately 2.5 million people exposed to more than 5 µg m^−3^ PM10 of desert dust origins [15]. According to Conte et al. [16], in the city of Lecce (south-eastern Italy), Saharan dust events contributed, with 20 µg m^−3^ on average, to ambient PM10 levels for a mean duration of 2.6 days in a seven-year period. In the metropolitan area of Rome, in central Italy, Saharan dust dry deposition occurred for approximately 22% of the days in a three-year period, with a contribution of approximately 10 µg m^−3^ on average to ambient PM10 levels [17].

Wang et al. [18] estimated the impact of dust exposure on health in central and western Europe based on the GEOS-Chem (http://www.geos-chem.org (accessed on20 September 2022)) global transport model; according to these authors, Italy ranks second among the European countries with the highest level of population-weighted averaged concentration exposure to total dust, with a value of 6.2 µg m^−3^, resulting in 27% of the total number of deaths related to the exposure to ambient PM10 levels in a two-year period (2016–2017).

Inhalation exposure entails the intake of any types of airborne substances by a receptor via the inhalation route in terms of magnitude, frequency, and duration [19]. Human exposure to mineral dust may have both short- and long-term impacts on health [20,21], with the frequency and intensity of the dust events playing a role in the occurrence of adverse health outcomes. Exposure to PM of Saharan origins, and hence the potential adverse health outcomes, depends on the diameter of PM particles. For example, PM < 10 µm (PM10) in diameter enters the respiratory tract, and finer particles may overcome the defences of the respiratory system and enter the bloodstream [22]. Gini et al. [23] estimated higher mass deposition of PM in the upper (approximately 80%) and lower (approximately 27%) respiratory tracts for the days affected by Saharan dust intrusions compared to not affected days. However, in the work by Alessandrini et al. [24] regarding the effects of different sized bins of Saharan dust on emergency hospitalisations in Rome, during dust-affected days emergency hospitalisations were associated with different fractions of PM (i.e., particles with diameters in the range of 2.5–10 and <10 µm) levels, whereas PM2.5 exerted no specific effect on morbidity. Moreover, based on a literature review, Karanasiou et al. [25] observed no significant association between PM2.5 and total or cause-specific daily mortality during Saharan dust outbreaks. The recent World Health Organization (WHO) [26] air quality guideline set the target of 24-h averaged concentration of 15 µg m^−3^ for short term exposure to ambient PM2.5 (i.e., 3–4 exceedance days per year), but no concentration levels are provided for PM originating from dust intrusions. However, best practices are provided by the WHO [26] to limit short-term exposure to the peaks of ambient PM2.5 levels during dust events. Regarding daily dust forecasts at regional scale, the website of the northern Africa–Middle East–Europe regional centre (https://sds-was.aemet.es, accessed on 11 October 2022) of the World Meteorological Organization (WMO) provides maps of dust surface concentration and dust optical depth with a time window of 72 h based on ensemble forecast obtained by interpolating the output of different prediction models. A dust monitoring and alert system has been operational in Spain and Portugal since 2001; this alert system sends daily forecasts to the involved air quality stakeholders based on (i) air-mass back trajectories and (ii) daily information provided by the WMO northern Africa-Middle East-Europe regional centre [27].

Based on a literature review, Tobías and Stafoggia [28] pointed out that binary (dust versus non-dust days) and continuous (portion of the bulk PM of dust origins) metrics are the methods used for dust exposure assessments in epidemiologic short-term health effects studies. For exposure assessment, the spatial and temporal variability of ambient concentrations can be better represented with CTMs (such as WRF-Chem; [29]) compared to fixed point measurements from monitoring stations [30].

The use of geographic information systems (GISs) dates back to the early 1990s, for a variety of applications in exposure assessment for air pollution epidemiology studies [31], including techniques for (i) evaluating spatial patterns of ambient concentrations based on geographically sparse air quality monitoring stations (e.g., land use regression modelling) [32], (ii) locating the population with respect to known emission sources (e.g., buffering), or (iii) defining exposure at a personal level with temporal and spatial resolution for a subgroup of receptors (modelling of time–activity patterns) [33].

For the population environmental exposure applications, there is the need to combine the spatial distribution of both population and surface dust concentration. To this end, Badaloni et al. [34] provided a resident population map in ESRI shapefile (.shp) format [35] with a spatial resolution of 1 km^2^ over a domain covering Italy.

Environmental protection and other governmental agencies may find useful the employment of an open-source risk prediction system based on the WRF-Chem model coupled with the QGIS software. This would allow the management of risks for the population posed by extreme pollution events and may also be useful in the case of natural hazards, including severe meteorological phenomena (e.g., heat waves, extreme precipitation, flooding, etc.).

The aim of the present paper is to estimate the contribution of desert dust to surface PM2.5 levels and population exposure during a sequence of dust intrusions from Sahara to the central Mediterranean in June 2021. This dust event has been recently investigated in the work by Rizza et al. [36], focusing on the influences of dust aerosol on the surface energy budget by altering radiation and cloud properties. In the present paper, this sequence of dust intrusions is simulated with the WRF-Chem numerical model and compared with spaceborne aerosol observations derived from MODIS [37], and with MERRA-2 [38] reanalysis. To evaluate short-term exposure to the dust intrusions, WRF-Chem simulations and the Italian resident population data are coupled by means of the open-source software QGIS [39]. This system represents a modern modelling approach for the development of quantitative analyses to improve hazard assessment and risk mitigation.

The remainder of this paper is organized as follows: Section 2 describes the numerical model configuration, the QGIS package, and the experimental data, including dust PM2.5 from MERRA-2 reanalysis, AOD from spaceborne observations, and the resident population map of Italy; in Section 3, results and discussions are presented; and finally, in Section 4, the conclusions of the study are drawn.

## 2. Materials and Methods

### 2.1. Description of Experimental Data

#### 2.1.1. Aerosol Optical Depth Derived by MODIS

The MODIS instrument has been on board the Terra and Aqua NASA spacecrafts since December 1999 and May 2002, respectively.

The two spacecrafts observe the entire Earth’s surface every 1 to 2 days, acquiring data in 36 spectral bands, or groups of wavelengths, spanning from the visible to the infrared, with high spatial resolution and near-daily global coverage (https://modis.gsfc.nasa.gov/about/, accessed on 16 June 2022). Aerosol characterization represents the core of the MODIS mission, and the AOD is still the most robust aerosol physical parameter derived from space [40].

The most recent collection (C006) of MODIS-AOD data is elaborated as a time averaged map of combined Dark Target (DT) [40] and Deep Blue (DB) [41] AOD at 550 nm, and is valid for both land and ocean surfaces [42]. This collection of MODIS AOD data provides a single AOD product combining both the DT and the DB AOD retrievals, and it is considered the “best-of” product for most quantitative purposes [42].

#### 2.1.2. Surface Dust PM2.5 Mass Concentrations Based on MERRA-2

MERRA-2 is the most recent global atmospheric reanalysis produced by the NASA Global Modeling and Assimilation Office using the Goddard Earth Observing System Model version 5.12.4.

The data collection denominated M2T1NXAER [43] consists of hourly time-averaged two-dimensional data in MERRA-2 of assimilated aerosol diagnostics, including surface mass concentration of aerosol components.

MERRA-2 data are downloaded from the GIOVANNI web-portal (https://giovanni.gsfc.nasa.gov/giovanni/, accessed on 8 June 2022), which provides time averaged maps of hourly dust surface mass concentration and PM2.5 dust with a spatial resolution of 0.5° × 0.625° [38] in the netCDF format [44].

#### 2.1.3. Geographical Information System and Population Map

For this study, we considered the map of population distributed over 1 km^2^ grid cells covering Italy based on the 2011 population census [34]. To obtain the population map distributed on cells, Badaloni et al. [34] calculated the population density per square meters for each census section by (i) assuming a homogeneous distribution of population, (ii) intersecting the grid cells and the census demographic data map, and (iii) summing the inter-section contributions related to each cell.

There is a time lag of approximately 10 years between the population map by Badaloni et al. [34] based on the 2011 population census by the Italian National Institute of Statistics (ISTAT, https://www.istat.it/en/ accessed on 2 September 2022) and the time (June 2021) when the sequence of dust intrusions occurred. To consider this variation in the amount of resident population, we scaled, for each region (i.e., level 2 of NUTS, the nomenclature of territorial units for statistics of the European Union), the 2011 population data with the 2021 total population data available from ISTAT (https://www.istat.it/en/ accessed on 2 September 2022) according to the method described in Soares et al. [45]:(1)$${SP}_{i,j}=P_{i}\times \frac{P_{j}^{*}}{P_{j}}$$
where ${SP}_{i,j}$ [unitless] is the scaled population in the *i*-th grid cell of the *j*-region in the population map; $P_{i}$ [unitless] is the 2011 population in the *i*-th grid cell in the population map; $P_{j}^{*}$ [unitless] is the total population according to ISTAT for 2021 for the *j*-region; $P_{j}$ [unitless] is the 2011 total population of the *j*-region according to the 2011 population census (ISTAT 2011). Table A1 shows $\frac{P_{j}^{*}}{P_{j}}$ values for each region grouped according to the Italian macro-regions (north, central, south and insular Italy) that are reported in Figure A1.

Finally, to evaluate the short-term exposure, QGIS assigns, in each grid cell of the population map, the time averaged (8-day) PM2.5 dust concentrations, which are calculated with the WRF-Chem model. Specifically, the coordinates of the centroid of each cell are considered for extracting the value of PM2.5 dust concentration from the netCDF file. Therefore, the predicted concentration value in the centroid of each 1 km^2^ cell is assumed to be representative of the respective cell of the population map.

### 2.2. QGIS Software

QGIS is an open-source multiplatform GIS with a graphical user interface, with common functions and features providing a spatial file browser and server/web applications. Moreover, the plugin architecture allows the use of additional features and functions providing extensibility and flexibility without altering the robustness of its core system. QGIS is useful for GIS data viewing and capturing, advanced GIS analysis, and presentations in the form of sophisticated maps, atlases, and reports. It supports a broad range of raster and vector data formats (https://www.qgis.org/en/site/about/index.html, accessed on 9 September 2022).

### 2.3. WRF-Chem Setup

The WRF-Chem model (version 4.2.1) has been utilized in a domain covering the northern Sahara Desert and the central Mediterranean, with 320 × 300 points and a horizontal grid spacing of 10 km. The initial and boundary conditions are provided by NCEP-FNL (Final) operational global analysis/forecast fields (0.25°) [46]. The simulation started on 17 June 2021 (00:00 UTC) and ended on 25 June 2021 (00:00 UTC).

The parameterizations of the physics package are reported in Table 1; in particular, the Yonsei University Scheme parameterization [47] is used to describe the planetary boundary-layer together with the revised MM5 surface layer similarity scheme [48], while the Unified Noah Land Surface Model is selected to represent the land surface processes [49]. The Four-Dimensional Data Assimilation scheme based on the Analysis Nudging [50] is adopted to consider the large-scale forcing on the numerical domain [51].

The radiative schemes are parameterized using the Rapid Radiative Transfer Model [52] for both short-wave (ra_sw_physics = 4) and long-wave (ra_lw_physics = 4), which are coupled with the Goddard Chemistry Aerosol Radiation and Transport (GOCART) model [53]. Concerning the microphysics parameterization, the two-moments aerosol-aware Thompson scheme has been used with prognostic variables for mixing ratios and number concentrations of the following five water species: cloud water, cloud ice, snow, rain, and a mixed hail-graupel class [54]. This aerosol-aware scheme is coupled with the GOCART aerosol model, enabling WRF-Chem to online simulate the effect of dust aerosol in the ice nucleation processes [55].

The chemistry/aerosol package is based on the GOCART model. It consists of seven bulk aerosol species, namely organic carbon, black carbon, other GOCART primary species (PM2.5, and PM10), sulphate (only secondary aerosol species), and three aerosol sectional species, namely mineral dust, sea spray, and volcanic ash.

The dust emission scheme designed by the Air Force Weather Agency (AFWA) is utilized and selected by the namelist variable dust_opt = 3 [56]. It considers five dust size bins, with a mean effective radius of 0.55, 1.4, 2.4, 4.5, and 8 μm. A detailed description of this emission scheme is provided by LeGrand et al. [56], Ukhov et al. [57], and Rizza et al. [36,58]. It is important to remark that, in this parameterization, the dust emission is controlled by the saltation of larger particles (50–100 μm) that are triggered by the wind shear at the surface, leading to the emission of smaller particles with a radius in the range of 1–10 μm by saltation bombardment.

### 2.4. QGIS Post-Processing of the WRF-Chem Output

Figure 1 summarizes the two-step post-processing procedure for the WRF-Chem output and population map file in the QGIS environment.

The coupling between the WRF-Chem output and QGIS is realized by interpolating data from the WRF native curvilinear grid to the rectilinear QGIS grid.

We used the QGIS data viewing properties to visualize and georeference the WRF-Chem output in netCDF format and the updated resident population map. Considering the QGIS functions dedicated to the analysis of vector and raster files, these two layers were superimposed, and the attributes merged into a new layer for linking the resident population to PM2.5 concentrations in each grid cell.

## 3. Results and Discussions

### 3.1. Synoptic Analysis and Event Description

The synoptic characteristics are studied throughout the final NCEP-FNL (Final) operational global analysis/forecast fields (0.25°), the same as those adopted for the WRF-Chem initial and boundary conditions.

The analysis clearly revealed a stationary synoptic configuration for the whole study period (from 17 to 24 June 2021). To show this, and for the sake of conciseness, we report in Figure 2, the averaged geopotential height (top panel) and the averaged wind (bottom panel) at 700 hPa, obtained considering all the available times (00, 06, 12, and 18 UTC) for the whole study period.

The map reveals the presence of two upper-level troughs over western and eastern Europe, separated by a wide upper-level ridge extending from north Africa up to the southern Mediterranean (Figure 2a). This configuration identifies a typical blocking pattern, in which the ‘omega’ structure remains stationary for a long period, facilitating the flowing of air masses from Africa to large European areas. Moderate-to-high zonal mid/upper-level winds, transporting warmer air masses, reach the Mediterranean, deflected by the ridge and persistently hitting the Italian peninsula, as shown in Figure 2b. This omega-like pattern is a typical configuration that favours the predominance in central and southern Italy of south westerly currents from Africa [6] and the corresponding intrusion of dust plumes from the Sahara Desert.

This is confirmed by analysing Figure 3a, which reports the average spatial pattern of the AOD from Aqua-MODIS retrievals [37]. Figure 3a clearly shows the presence of dust aerosol in the central Mediterranean basin, with a relatively high value of average AOD (~1). This means that, during the investigated period (17–24 June 2021), a sequence of intense dust plumes has been transported from the Sahara toward central and southern Italy.

### 3.2. Evaluation of Aerosol Optical Depth Based on WRF-Chem Output and Comparison with MODIS Retrievals

Before starting with the analysis of the exposure, it is important to evaluate how the model reproduces the dust amount in the numerical domain. This is usually done by considering (i) the amount of dust in the whole atmospheric column and (ii) the concentration of PM at ground level. AOD is an optical quantity that is specifically designed to calculate the total aerosol loading (both of natural and anthropogenic origins). In WRF-Chem, this quantity is calculated off-line by integrating, in the vertical, the gridded aerosol extinction coefficient at 550 nm (EXT55, km^−1^) according to the Maxwell–Garnett [59] mixing rule (aer_op_opt = 2):(2)$$AOD\left(x,y\right)=\int_{z_{0}}^{z_{h}}EXT55\left(x,y,z\right)dz$$
where $z_{0}$ is the first vertical level, $z_{h}$ is equal to 20 km, and dz represents the gradient of the vertical layers.

The spacecraft Aqua, whose data are used here, overpasses the Equator at 13:30 LT. The time-average of Equation (2) is calculated by considering only the time windows corresponding to the daily MODIS-Aqua overpasses over Italy (≈13:00 UTC).

In Figure 3, we report the time averaged AOD from the WRF-Chem model (Figure 3b) and the combined DT-DB products of MODIS-Aqua (Figure 3a). Simulated AOD values are generally overestimated over Algeria (northern Africa) and underestimated over the central Mediterranean basin and the Italian peninsula. Calculating the spatial average of the time averaged (17–24 June) AOD, in an inner box domain (denoted by the squared dotted lines in Figure 3) of coordinates (lat, lon) = (37:47, 7:17), we obtained AOD values of 0.46 for MODIS and 0.37 for WRF-Chem.

This is likely related to the difficulty in retrieving the AOD over brighter surfaces, for which the “Deep Blue” algorithm [41] has been developed, but it is still well-known for its underestimation over the Sahara [60]. The under-estimation over the Mediterranean is rather due to neglecting the contribution of the sea spray aerosol particles to the total AOD.

### 3.3. Evaluation of the Time Average Surface Dust PM2.5 and Comparison with MERRA-2 Reanalysis

To calculate the exposure of the population, we first need to consider the spatial distribution of surface dust concentration expressed in terms of PM2.5. To this end, we refer to the MERRA-2 assimilated aerosol M2T1NXAER data collection, in particular to the “dust surface mass concentration—PM2.5, time average” variable, downloaded from the Giovanni NASA web-portal.

The outputs of MERRA-2 are provided at a (lat, lon) resolution of 0.5° × 0.625°, so they need to be “re-gridded” before being compared with WRF-Chem outputs, which are provided at 10 × 10 km^2^ resolution. This is done with the ESMF_regrid function, which is part of a suite of re-gridding routines based on the Earth System Modeling Framework (ESMF; http://earthsystemmodeling.org accessed on 12 September 2022).

The results of the re-gridding on the time-averaged period (17–24 June) are shown in Figure 4, which reports both WRF-Chem (Figure 4a) and MERRA-2 (Figure 4b) surface dust PM2.5 concentrations. It is quite evident that WRF-Chem well reproduced the surface PM2.5 geographical pattern depicted by MERRA-2 reanalysis, with higher concentrations of PM2.5 over central and southern Italy compared to northern Italy. Calculating the spatial average of the time-averaged surface dust PM2.5 concentrations, a general underestimation of the WRF-Chem simulations is obtained (18.1 µg m^−3^) with respect to MERRA-2 reanalysis (24.2 µg m^−3^).

### 3.4. Exposure Evaluation at the National and Regional Scale

To consider the impact of the sequence of Saharan dust intrusions on air quality, the population exposure was evaluated for the Italian macro-regions (Table A1) and the whole of Italy.

Figure 5 shows the dust PM2.5 exposure of the resident population of Italy and its macro-regions, grouped according to exposure classes of 5 µg m^−3^ intervals in the range of 0–35 µg m^−3^ and determined by elaborating the WRF-Chem output on the QGIS-based population map. The exposure classes are split into 5 µg m^−3^ intervals to let the reader easily compare the recommended daily air quality levels for ambient PM2.5 concentrations [26] with the 8-day averaged dust PM2.5 concentration.

At the national level, during the sequence of dust events, the highest percentage (38%) of resident population was exposed to averaged PM2.5 concentrations up to 5 µg m^−3^, followed by exposure classes in the ranges 15–20 and 20–25 µg m^−3^ dust PM2.5, affecting approximately 17 and 13% of the resident population, respectively (Figure 5a). More than 4% of the resident population of Italy had been exposed to eight-day averaged PM2.5 concentrations > 30 µg m^−3^ for a total amount of approximately 2.5 million people (Figure 5a and Figure A2f).

Considering the Italian macro-regions, no residents had been exposed to dust PM2.5 concentration > 15 µg m^−3^ in north Italy (Figure 5b and Figure A2c–f), whereas approximately 28% (5.6 million people approximately) and more than 38% (approximately 4.6 million people) of the resident population of south/insular Italy (Figure 5d), and central Italy (Figure 5c), respectively, had been exposed to eight-day averaged concentration in the range of 15–20 µg m^−3^ (Figure A2c).

As a general tendency, as is easily noticeable from Figure 5, it can be stated how in northern Italy most of the resident population (approximately 83.2%; 22.5 million people) was exposed to relatively low PM2.5 concentrations during the dust sequence (up to 5 µg m^−3^), while the opposite was found for the regions of southern/insular and central Italy.

For short term exposure assessment, time-varying factors should be considered, as exposure only related to the place of residence does not consider population dynamics [33] such as the rate of time spent in indoor vs. outdoor activities. However, the time activity patterns of the study population are generally considered in studies related to spatially heterogeneous airborne pollutants from local pollution sources (e.g., NOx from traffic emissions) and restricted domain (e.g., urban scale), whereas transboundary dust transport involves broader domains and hence study populations.

Because of the risk posed by exposure to PM2.5 of dust origins, the WHO lists good practices aimed at limiting the population exposure to peaks in ambient PM2.5 levels originating from dust events, including early warning systems and dust forecasting programmes [26]. To this end, the present methodology can be useful if incorporated into a forecast system that includes a CTM. In this context, it could deliver early-warning alarm messages to the regions with the most exposed population to severe pollution episodes. A dust alert system is useful for informing susceptible populations of the need for reducing or avoiding outdoor activities, and for adopting exceptional and temporary measures aimed at decreasing local emission sources [27]. Future studies may consider the variation in exposure of resident population at finer administrative levels (e.g., municipalities (NUTS3) for the regions with considerable extension or particular orography), or age classes.

## 4. Conclusions

A sequence of dust intrusions, from the Sahara Desert, affected the central Mediterranean from 17 to 24 June 2021. This event was characterized by an eight-day averaged value of aerosol optical depth (AOD) up to approximately 1 and by a typical omega-like synoptic pattern that favoured the predominance, in central and southern Italy, of south-westerly currents from Africa.

The population exposure was analysed by coupling the WRF-Chem model and the open-source QGIS, for combining the predicted PM2.5 surface dust concentration with a resident population map of Italy.

WRF-Chem outputs were firstly compared with spaceborne aerosol observations derived from MODIS and with MERRA-2 reanalysis for PM2.5 surface dust concentration. Comparisons showed an underestimation of simulated compared to observed AOD of 0.37 for WRF-Chem and 0.46 for MODIS. Considering the surface dust PM2.5 concentrations, WRF-Chem well reproduces the geographical pattern depicted by the MERRA-2 reanalysis, even if with an average under-prediction of approximately 25%.

The population exposure was evaluated for Italy and its macro-regions (i.e., north, central, and south and insular Italy). The resident population was grouped according to exposure classes of 5 µg m^−3^ intervals in the range of 0–35 µg m^−3^ to allow the reader to easily compare the recommended daily air quality levels for ambient PM2.5 concentrations by the World Health Organization with the eight-day averaged dust PM2.5 concentration.

Results showed that, at the national level, the exposure class of up to 5 µg m^−3^ dust PM2.5 had the highest percentage (38%) of resident population, followed by exposure classes in the range of 15–25 µg m^−3^. Moreover, approximately 4.2% (2.5 million people) of the resident population of Italy had been exposed to eight-day averaged PM2.5 concentrations >30 µg m^−3^.

The comparison of exposure classes calculated for Italy and its macro-regions shows that the exposure to this dust sequence varies with the location and entity of the resident population, with most of the population of north Italy in the lowest class of exposure (i.e., 5 µg m^−3^), and more than a half (54–57%) of the population of central, south and insular Italy in the range of 15–25 µg m^−3^.

Future studies shall consider the variation in exposure of resident population at finer administrative levels, or age classes.

As seen in this work, the areas affected by Saharan dust vary according to the specific events, which are in turn linked to peculiar atmospheric conditions. The use of an integrated tool such as the one proposed, based on meteorological and chemical modelling, allows one to predict the areas potentially most exposed to the risks associated with extreme pollution events and/or natural hazards. The most precise possible knowledge of the areas affected by PM2.5 intrusions (or, more generally, by dust or other pollutants), combined with the information of the zones with the highest resident population density, allows one to obtain localized information and to facilitate the planning of appropriate prevention measures for these high-risks areas. In this context, QGIS postprocessing is a valuable solution for geolocation and accurate spatial data analysis.

## Figures and Tables

**Figure 1 ijerph-20-05598-f001:**
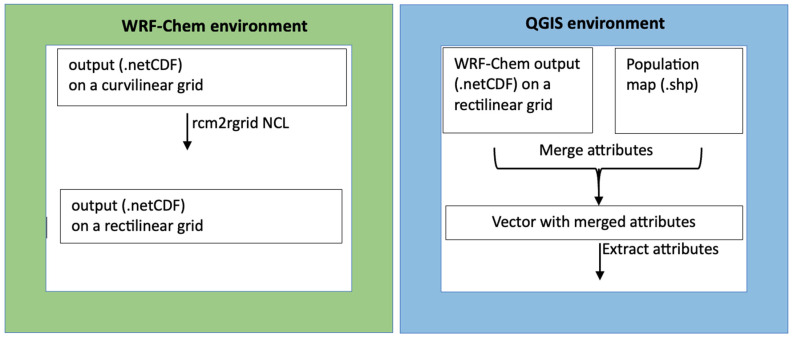
Summary scheme for the merging of WRF-Chem output and population map file in QGIS environment.

**Figure 2 ijerph-20-05598-f002:**
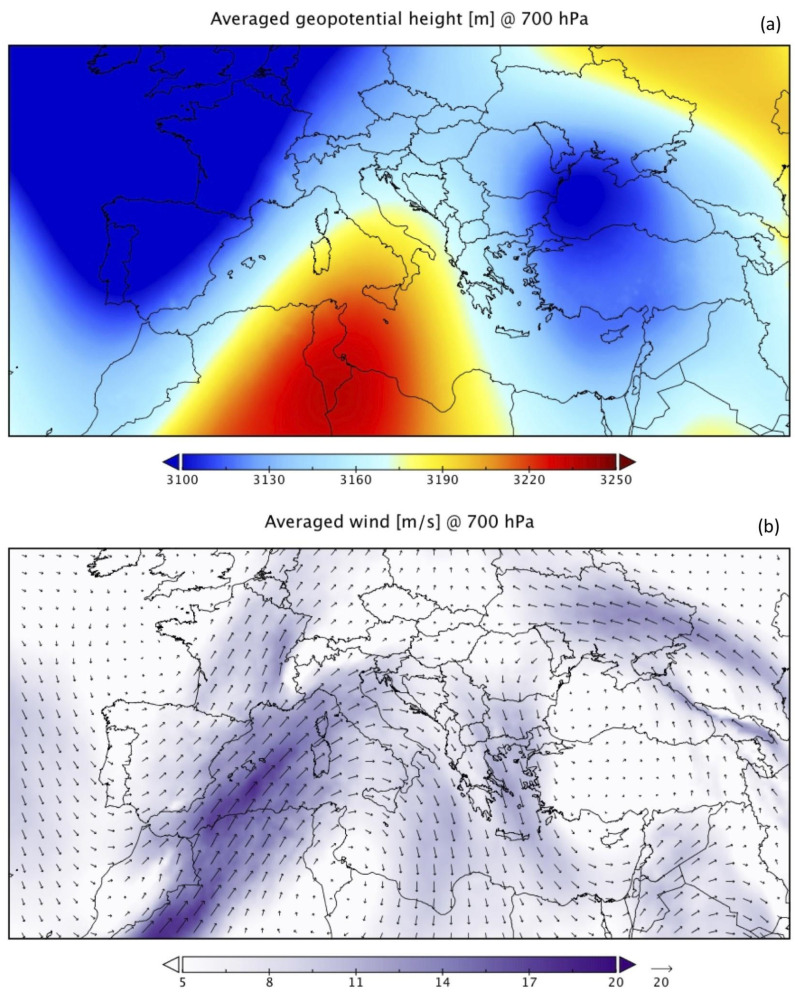
Full-period (17–24 June) averaged geopotential height (m) (**a**), and wind (vectors and speed; m s^−1^) (**b**) at 700 hPa.

**Figure 3 ijerph-20-05598-f003:**
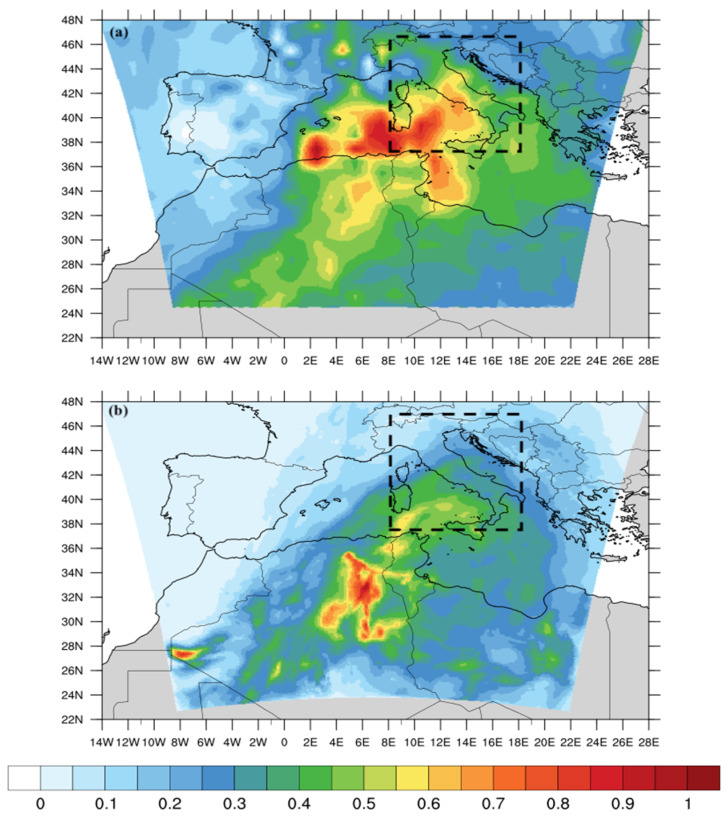
Full-period (17–24 June) averaged AOD from (**a**) MODIS-Aqua combined (DT + DB) algorithms and (**b**) WRF-Chem model. Spatial average is calculated in the squared dotted box.

**Figure 4 ijerph-20-05598-f004:**
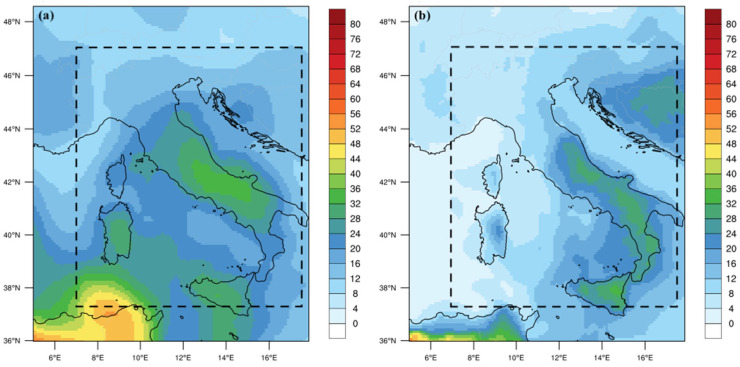
Comparison between full-period (17–24 June) averaged maps of surface dust PM2.5 concentrations from (**a**) MERRA-2 reanalysis, and (**b**) WRF-Chem simulation. Spatial average is calculated in the squared dotted box. Units are µg m^−3^.

**Figure 5 ijerph-20-05598-f005:**
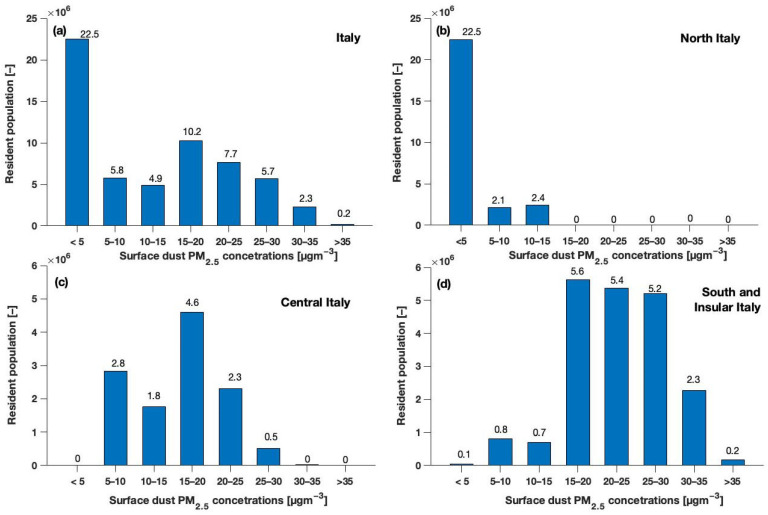
Classes of exposure to the full period (17–24 June) averaged surface dust PM2.5 mass concentrations simulated with WRF-Chem for the resident population in (**a**) Italy, and in the Italian macro-regions, namely (**b**) north, (**c**) central, and (**d**) south and insular Italy.

**Table 1 ijerph-20-05598-t001:** Physical and chemical options and the namelist setting.

Microphysics	mp_physics = 28	Aerosol aware scheme
LW/SW radiation	ra_sw_physics = 4 ra_lw_physics = 4	RRTMG Shortwave and Longwave Schemes
Surface Layer	sf_sfclay_physics = 1	Revised MM5 Scheme
Planetary boundary layer	sf_pbl_physics = 1	Yonsei University Scheme
Land surface	sf_surface_physics = 2	Unified Noah Land Surface Model
Nudging	grid_fdda = 1	Analysis Nudging
chem_opt	chem_opt = 300	GOCART
dust_opt	dust_opt = 3	AFWA dust emission scheme

## Data Availability

Not applicable.

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
