# Peer review of "Exposure Assessment of Ambient PM2.5 Levels during a Sequence of Dust Episodes: A Case Study Coupling the WRF-Chem Model with GIS-Based Postprocessing"

_ijerph, 2023, doi:10.3390/ijerph20085598_

Round 1
Reviewer 1 Report
PM2.5 or PM10 with Higher concentration caused by Dust events affect human health, this issue has become focused research in environmental health and public health. The manuscript contributed an exposure assessment case is meaningful. The coupling scheme of WRF-Chem model with GIS is innovative and result showed that it is effective. Totaly, i think it can be accepted if the problems as followed are handled well.
1) In Figure 1. the map should be drawn in more detail and be consistent with the latter WRF-Chem map not only Range but projection.
2) In paragraph from line 302 to 313, Simulated AOD are generally overestimated over Algeria underestimated over the central Mediterranean basin and the Italian Peninsula, because MODIS AOD algorithm works bad over some places, how to decide which AOD you should take to analyse? Could you evaluate the two result from Ground observation such as AERONET if available?
3) Line 310, the word TYPO “well-know” should be “well-known”
4) Figure 5, concentrations,the research give that general underestimation of the WRF-Chem simulations (18.1 µgm-3 ) with respect to MERRA-2 reanalysis (24.2 µgm-3 ). which one is better? Why you select WRF-Chem simulation for finally application?Could you evaluate the two result from Ground observation like AOD?
5) Figure 6. What is the unit of horizontal axis in the Figure (a) and (b), do you share it with the next Figures? Plz explain or clear it.
Author Response
To Referee 1,
PM2.5 or PM10 with Higher concentration caused by Dust events affect human health, this issue has become focused research in environmental health and public health. The manuscript contributed an exposure assessment case is meaningful. The coupling scheme of WRF-Chem model with GIS is innovative and result showed that it is effective. Totaly, i think it can be accepted if the problems as followed are handled well.
We are grateful to the reviewer for the valuable comments and suggestions on the present paper. The modified parts of the text are highlighted in green colour in the revised manuscript. Please, find below our point-by-point answers (in red) to your comments.
1) In Figure 1. the map should be drawn in more detail and be consistent with the latter WRF-Chem map not only Range but projection.
In agreement to the remarks raised by both reviewers we removed Figure 1 from the revised manuscript recognizing that it does not provide useful additional information.
2) In paragraph from line 302 to 313, Simulated AOD are generally overestimated over Algeria underestimated over the central Mediterranean basin and the Italian Peninsula, because MODIS AOD algorithm works bad over some places, how to decide which AOD you should take to analyse? Could you evaluate the two results from Ground observation such as AERONET if available?
After a careful analysis of AERONET data we found that, for the considered period, AERONET does not have sufficient level 3 data. For this reason, we decided to proceed by utilising MODIS AOD for the comparison. It is important to remark that MODIS-AOD is generally considered the most robust product for AOD among of the whole set of multi sensor measurements (from ground and space).
3) Line 310, the word TYPO “well-know” should be “well-known”
We revised the text according to the reviewer’s suggestion.
4) Figure 5, concentrations, the research give that general underestimation of the WRF-Chem simulations (18.1 µgm-3 ) with respect to MERRA-2 reanalysis (24.2 µgm-3 ). which one is better? Why you select WRF-Chem simulation for finally application? Could you evaluate the two result from Ground observation like AOD?
WRF-Chem is one of the most widely used models for the study and forecast of air quality and meteorology-chemistry interactions at a regional scale. Furthermore, it is an open-source code available to community. In this context it is possible to configure it depending on the proper utilization. On the other side, MERRA-2 is a global model with a horizontal resolution of about 50-60 km. It is not open-source and consequently it is not possible to change the configuration/setup. Regarding ground observations, it is important to point out that MERRA-2 reanalyses incorporates several space-based observations of aerosols (also AERONET measurements) as well as many space-borne satellite products. In this context, MERRA-2 can be considered as a reference for comparisons with WRF-Chem simulations.
5) Figure 6. What is the unit of horizontal axis in the Figure (a) and (b), do you share it with the next Figures? Plz explain or clear it.
The four panels of figure 6 share the same units for the respective axes. To improve the readability of Figure 6, we added titles and units to the x and y axes of (a) and (b) panels.

Reviewer 2 Report
A. At end of introduction, please include text which succinctly states the objectives / hypothesis for this paper.
B. Can this approach to integration of QGIS with WRF CHEM be easily adapted by other authors? If so how? Can an open source contribution become part of this paper with instructions how to implement? If so, the work may become a very popular paper to site as it would be enabling for other researcher.
C. How oft does the MODIS provide you with data? Is this rate sufficient for the study?
D. What is advantage of integrating GIS with WRF rather than MERRA-2 data?
E. Is Fig. 1 in any way useful? Seems unnecessary in current form. Can other information be provided here?
F. FIg 6 - any way to show a map or graphical presentation of data? It could be useful to develop early warning system for areas of high exposure.
G. The main advance of this paper is coupling of the model and data to the GIS. There is little other new data warranting publication. So it's important authors describe this more and explain how the novel GIS mapping informs and plays important role to describe human exposure. Too may model technical details and not enough new content.
H. On page 12 authors failed to complete several required sections after conclusion.
I. Its not clear what table A1 is crucial to the paper given human exposure and its consequences was not a major theme in text.
Author Response
To Referee 2,
We are grateful to the reviewer for the valuable comments and suggestions on the present paper. The modified parts of the text are highlighted in green colour in the revised manuscript. Please, find below our point-by-point answers (in red) to your comments.
- At end of introduction, please include text which succinctly states the objectives / hypothesis for this paper.
The main objectives of the study were already reported in the Introduction section. In order to follow the reviewer’s suggestion, part of the comment has been moved at the end of the Introduction, just before the description of the paper's organization.
- Can this approach to integration of QGIS with WRF CHEM be easily adapted by other authors? If so how? Can an open source contribution become part of this paper with instructions how to implement? If so, the work may become a very popular paper to site as it would be enabling for other researcher.
The proposed approach can be surely adapted/replicated. The main steps of the post-processing procedure were already described in paragraph 2.4 and summarized in Figure 1. Future works will consider the development of a complete QGIS plugin (not ready yet) to allow the automation of the steps reported in the above-mentioned paragraph.
- How oft does the MODIS provide you with data? Is this rate sufficient for the study?
MODIS sensor is onboard the Aqua and Terra polar orbiting spacecrafts. These satellites scan the same area on Earth with a time lag of about three hours. The entire Earth's surface is scanned every one to two days. MODIS-AOD data are accessible as level 3 (i.e., level 2 data, properly georeferenced, and temporally averaged) through the GIOVANNI-NASA web portal. These temporal-average data are usually utilized in air-quality study when the Aerosol Optical Depth is utilized as proxy for aerosol impact on air quality over the whole vertical column.
- What is advantage of integrating GIS with WRF rather than MERRA-2 data?
This is a relevant point, also raised by the Reviewer n.1. WRF-Chem is one of the most widely used models for to study and forecast of air quality and meteorology-chemistry interactions at a regional scale. Furthermore, it is an open-source code that is available to community. In this context it is possible to configure it depending on the proper utilization. On the other side, MERRA-2 is a global model with a horizontal resolution of about 50-60 km (coarser resolution compared to WRF). It is not open-source and consequently it is not possible to change the configuration/setup.
- Is Fig. 1 in any way useful?Seems unnecessary in current form. Can other information be provided here?
We agree with the reviewer about Figure 1. Therefore, we removed the figure in the revised manuscript because it does not provide particularly useful information.
- Fig 6 - any way to show a map or graphical presentation of data?It could be useful to develop early warning system for areas of high exposure.
We agree with the reviewer that showing graphical results could be more intuitive and useful for the development of an early warning system. To this purpose we added the following Figure A2 to the Appendix section to provide a geographical distribution of the classes of exposure that are shown in Figure 5a for the whole Italy. We think that the histogram representation is informative as well because it provides fundamental information for exposure assessments, that is how much population is subject to a given concentration. Therefore, the future early warning system should rely both on presentation of data with a map and a bar-chart as the one in Figure 5.
Figure A2. Geographical distribution of the classes of exposure to the full period (17–24 June) averaged surface dust PM2.5 mass concentrations simulated with WRF-Chem for the resident population in Italy, namely the exposure class of (a) <10 µgm-3 for 28.3 million residents, (b) 10–15 µgm-3 for 4.9 million residents, (c) 15–20 µgm-3 for 10.2 million residents, (d) 20–25 µgm-3 for 7.7 million residents, (e) 25–30 µgm-3 for 5.7 million residents, and (f) >30 µgm-3 for 2.5 million residents.
- The main advance of this paper is coupling of the model and data to the GIS.There is little other new data warranting publication. So it's important authors describe this more and explain how the novel GIS mapping informs and plays important role to describe human exposure. Too many model technical details and not enough new content.
We agree with the reviewer, it is anyway important to point out that this technique is already described in chapters 2.1.3, 2.2, 2.4 and results discussed on 3.4. Moreover, figure 1 show the flow diagram describing the scheme for the merging of WRF-Chem output and population map file in QGIS environment. This description is primarily addressed to GIS experts so in this context we believe there is no needing of further details. We added the following sentence (green colour) to the Conclusions section:
‘’ The most precise possible knowledge of the areas affected by PM2.5 intrusions … combined with the information of the zones with the highest resident population density, allows to obtain localized information and to facilitate the planning of appropriate prevention measures for these high-risks areas. In this context, QGIS postprocessing is a valuable solution for geolocation and accurate spatial data analysis.’’
- On page 12 authors failed to complete several required sections after conclusion.
We revised the required sections according to your suggestion.
- Its not clear what table A1 is crucial to the paper given human exposure and its consequences was not a major theme in text.
The purpose of Table A1 is to provide supplemental information able to facilitate readers to understand the correction performed on the population map by Equation 1.
Specifically, Table A1 is reported in the Appendix section to better identify the Italian regions whit higher [(Pj*/Pj) >1] and lower [(Pj*/Pj) <1] resident population in 2021 compared to 2011.
